# Body Composition Profiles of Applicants to a Physical Education and Sports Major in Southeastern Mexico

**DOI:** 10.3390/ijerph192315685

**Published:** 2022-11-25

**Authors:** Edgar I. Gasperín-Rodríguez, Julio A. Gómez-Figueroa, Luis M. Gómez-Miranda, Paul T. Ríos-Gallardo, Carolina Palmeros-Exsome, Marco A. Hernández-Lepe, José Moncada-Jiménez, Diego A. Bonilla

**Affiliations:** 1Nutrition Faculty, Veracruzan University, Veracruz 91700, Mexico; 2Physical Education, Sport and Recreation School, Veracruzan University, Veracruz 94294, Mexico; 3Sports Faculty, Autonomous University of Baja California, Tijuana 22390, Mexico; 4Research Group UABC-CA-341 in “Physical Performance and Health”, Autonomous University of Baja California, Tijuana 22390, Mexico; 5Medical and Psychology School, Autonomous University of Baja California, Tijuana 22390, Mexico; 6Human Movement Sciences Research Center (CIMOHU), University of Costa Rica, San José 11501, Costa Rica; 7Research Division, Dynamical Business & Science Society–DBSS International SAS, Bogotá 110311, Colombia; 8Research Group in Physical Activity, Sports and Health Sciences—GICAFS, Universidad de Córdoba, Montería 230002, Colombia; 9Research Group in Biochemistry and Molecular Biology, Faculty of Sciences and Education, Universidad Distrital Francisco José de Caldas, Bogotá 110311, Colombia; 10Sport Genomics Research Group, Department of Genetics, Physical Anthropology and Animal Physiology, Faculty of Science and Technology, University of the Basque Country (UPV/EHU), 48940 Leioa, Spain

**Keywords:** body fat, public health students, physical education and sports major, university health services, unsupervised machine learning

## Abstract

This study aimed to determine the body composition profile of candidates applying for a Physical Education and Sports major. 327 young adults (F: 87, M: 240) participated in this cross-sectional study. Nutritional status and body composition analysis were performed, and the profiles were generated using an unsupervised machine learning algorithm. Body mass index (BMI), percentage of fat mass (%FM), percentage of muscle mass (%MM), metabolic age (MA), basal metabolic rate (BMR), and visceral fat level (VFL) were used as input variables. BMI values were normal-weight although VFL was significantly higher in men (<0.001; η^2^ = 0.104). MA was positively correlated with BMR (0.81 [0.77, 0.85]; *p* < 0.01), BMI (0.87 [0.84, 0.90]; *p* < 0.01), and VFL (0.77 [0.72, 0.81]; *p* < 0.01). The hierarchical clustering analysis revealed two significantly different age-independent profiles: Cluster 1 (n = 265), applicants of both sexes that were shorter, lighter, with lower adiposity and higher lean mass; and, Cluster 2 (n = 62), a group of overweight male applicants with higher VFL, taller, with lower %MM and estimated energy expended at rest. We identified two profiles that might help universities, counselors and teachers/lecturers to identify applicants in which is necessary to increase physical activity levels and improve dietary habits.

## 1. Introduction

The increase in sedentary behavior (e.g., smartphone addiction or avoidance of physical activity tasks) represents one of the main problems for public health in today’s society. Freshman or first-year college students are a group at potential risk of chronic non-communicable diseases [1] since the onset of undergraduate activities involves changes in lifestyle, family distancing, personal independence, and stressful situations [2]. This group of young adults is characterized for unhealthy habits and reduced physical fitness which result in negative changes in body composition [3]. In the case of Physical Education (PES) majors, it is important to reach a balance between theory and practice in their academic development, although this this is not often perceived due to the high stress levels.

Body composition is defined as a branch of human biology dealing with the in vivo quantification of body components. Body composition is a key measure in biomedical sciences such as nutrition, medicine, anthropology, and physical education for its importance in determining nutritional status, both for health and disease [4]. The body composition is considered one of the factors that impact on general fitness [5]. The assessment of body composition consists on the compartmentalization of the total body mass in different components or models (e.g., two- or four-compartment model). These components suffer changes as a result of growth, development, dietary habits, exercise interventions, and ageing [6].

Currently, there is a wide diversity of methods used for the measurement of the body composition, which includes indirect and double indirect methods [7]. Anthropometry is one of the most frequently used techniques for assessing the nutritional status and body composition in several populations [8,9]. Although it is a common anthropometric indicator to diagnose obesity that is promoted by the World Health Organization (WHO), the body mass index (BMI) has certain misclassification problems given it does not differentiate the body composition of the individuals [10]. In this regard, alternative anthropometry-related data might be useful to identify an excess or to estimate whole-body fat [11]. On the other hand, bioelectrical impedance analysis (BIA) is another common method to assess body composition through impedance quantification (i.e., opposition that biological tissues present to alternating current at very low voltage intensity) [12]. BIA is a non-invasive and innocuous technique, with low intra- and inter-observer variability; however, it has a considerable estimation error depending on the equation used to estimate body composition. Hence, the bioelectrical impedance vector analysis (BIVA) is recommended for individual analyses [13]. For large populations, BIA has been widely used in different age and population groups [14].

Nowadays, universities offering majors in Health Sciences, including Physical Education, Sports, and Recreation, request the completion of a series of physical fitness tests from their candidates. Some of these tests focus only on physical performance (e.g., muscle strength test, power, speed) although others include variables such as body composition [15]. As a worldwide multi-factorial phenomenon, the obesogenic environment is also influenced by the transition period from high school to university where there is a decline in healthy behaviors, physical activity, and unhealthy body composition [16]. Some of the students’ common behaviors are the excessive consumption of calories, lack of sleep hygiene, and a high level of alcohol intake [17]. Latin American countries like Mexico have high-quality food available all year; however, unhealthy habits are common among the population which has raised concerns in health and physical activity practitioners to intervene before the overweight and obesity levels increase. In this sense, a significant increase in the prevalence of overweight and obesity in Mexican college students [18]. Since there is a lack of studies in Mexican candidates to a PES major, the aim of this study was to profile for the first time body composition of applicants to a PES major in a southeastern Mexican university. Based on previous findings in Mexican college students [19], we hypothesize that there will be a sex-dependent influence on the generated profiles.

## 2. Materials and Methods

### 2.1. Study Design

This was a cross-sectional study reported in accordance with the Strengthening the Reporting of Observational Studies in Epidemiology–STROBE guidelines [20].

### 2.2. Setting

The study was carried out from January to February 2019. This research was conducted in accordance with the ethical guidelines of the Declaration of Helsinki as an investigation project of the Physical Education, Sport and Recreation School at the Universidad Veracruzana (Veracruz, Mexico). The research protocol and informed consent were approved by the institutional review board (code: 0019385). The potentially eligible population was reached by email and posters at the University facilities.

### 2.3. Participants

All applicants to the Physical Education and Sports major at the Universidad Veracruzana were potentially eligible to participate in this study. The selection was based on the following inclusion criteria: (i) applicants to the Physical Education and Sports major; (ii) not have any medical restriction that would make it impossible to take anthropometric measurements; (iii) to sign the informed consent.

### 2.4. Variables

Stature (cm) and body mass (kg) were measured and the BMI was derived, subsequently. BIA-related data were also collected, including percentage of fat mass (%FM); percentage of muscle mass (%MM); metabolic age (MA, years); basal metabolic rate (BMR, kcal); and, visceral fat level (VFL, cm^3^).

### 2.5. Data Sources/Measurement

The body composition data were collected between 7:00–9:00 am (GMT-5). The participants were measured before eating food and after voiding their bladders and evacuated. This scheduling was especially relevant because body composition variables might be dramatically modified by recent physical activity. The assessments were made in the facilities of the Laboratory of Physical Performance of the Universidad Veracruzana. The measurement procedure was as follows: (i) applicants were organized in small groups and taken to the assessment room. In the assessment room, two staff members explained the measurement protocol to reduce the technical error of measurement; (ii) stature was measured by the researchers, and age and sex data were recorded; (iii) finally, the participants stepped on the BIA scale and body composition variables were recorded.

#### 2.5.1. Anthropometric Assessment

The anthropometric measurements were performed according to the International Standards for Anthropometric Assessment published by the International Society for the Advancement of Kineanthropometry (ISAK) [21]. The stature was measured with a one-millimeter graduation stadiometer (Seca Model 214, Seca Corp., Hanover, MD, USA). Each participant was instructed to stand barefoot with the back to the stadiometer and heels together. The head upright with the lower orbital rim in the same horizontal plane as the external auditory canal (Frankfurt plane).

Body mass was measured with a portable digital scale to the nearest 100-g (Omron HBF-510^®^, Omron Healthcare, Inc., Hoffman Estates, IL, USA). The participants were asked to stand still barefoot and after removing as many clothes as possible for this measurement. The BMI was calculated (kg·m^−2^) and interpreted as indicated by the WHO in normal (18.5–24.9 kg·m^−2^), overweight (25–29.9 kg·m^−2^), or obesity (>30 kg·m^−2^). The anthropometric measures were performed twice by certified Level 1 anthropometrists with a technical error of measurement of <1% which is considered acceptable by the ISAK recommendations.

#### 2.5.2. Bioelectrical Impedance Analysis

A hand-to-foot bioelectric monitor Omron HBF-510^®^ (Omron Healthcare, Inc., Hoffman Estates, IL, USA) was used for the BIA (50 kHz and <500 µA). The estimated values of %FM, %MM, MA, BMR, and VFL were recorded after asking the participants not to carry metallic objects. It should be noted that MA is an alternative index of BMR as well as an indicator of high-risk individuals for metabolic syndrome [22]. The Omron HBF-510^®^ has been previously shown to significantly correlate with BOD-POD measurements (r = 0.95, *p* = 0.001) when assessing %FM in male and female college students [23].

### 2.6. Sample Size

Non-probability convenience sampling was used. After the announcement to participate in this study, a total of 327 apparently healthy applicants between 18 and 19 years that signed the informed consent were considered as enrolled.

### 2.7. Statistical Analyses

Descriptive statistics was expressed as mean and standard deviation (95% CI). All data of the included variables were analyzed with the Mann-Whitney U test to determine differences between sexes and generated profiles. Eta-squared (η^2^) was used to report the magnitude of differences assuming a 0.09, 0.14, and >0.22 as small, medium, and large effect size [24]. These statistical calculations were performed using the IBM SPSS v26 (IBM Corp., Armonk, NY, USA) by establishing a significance level of *p* < 0.05. A clustering analysis was performed as previous studies [25,26] to generate the profiles based on all measured variables within the free software environment for statistical computing and graphics R v4.0.2 [27]. The number of clusters was determined using the ‘NbClust’ function testing criterion algorithms that compared the two- to eight-cluster solutions. The internal validation of clustering results for k-means, PAM, hierarchical and CLARA algorithms was performed with the package ‘clValid’ which implements an evaluation of the goodness of the resulting clusters by comparing Dunn, Silhouette and Connectivity criteria measures [28].

## 3. Results

Included participants were 240 males (73.4%) and 87 females (26.6%). As expected, significantly higher values with medium-to-large effect sizes were found in men on stature (η^2^ = 0.394), body mass (η^2^ = 0.185), %MM (η^2^ = 0.526), BMR (η^2^ = 0.549) when compared to women. On the contrary, female participants had a significantly higher %FM (η^2^ = 0.375) than men. Although the mean values of BMI were normal in males and females, there was a significant difference in VFL (men = 5.8 cm^3^ vs. women = 3.8 cm^3^; η^2^ = 0.104). Descriptive analysis by sex is shown in Table 1.

An exploratory analysis of variance indicated that there was no significant interaction between the age group and sex on body mass, BMI or %FM (*p* > 0.05). Likewise, neither differences between the age groups (*p* > 0.05) nor between sexes (*p* > 0.05). Therefore, the percentile norms are presented together for males and females aged 18 and 19 years old (Table 2).

Pearson’s correlations (r, 95% CI; *p* value) were performed to explore the relationship between variables (Figure 1). BM had significantly positive correlations with BMR (0.87 [0.84, 0.89]; *p* < 0.01) and VFL (0.81 [0.77, 0.84]; *p* < 0.01) while BMI was significantly correlated with VFL (0.88 [0.85, 0.90]; *p* < 0.01). As expected, a significantly negative correlation was found between %FM and %MM (−0.81 [−0.84, −0.77]; *p* < 0.01). This correlation analysis also showed that MA was significantly and positively correlated with BM (0.81 [0.77, 0.85]; *p* < 0.01), BMI (0.87 [0.84, 0.90]; *p* < 0.01), and VFL (0.77 [0.72, 0.81]; *p* < 0.01). See Appendix A for the complete correlation statistics results.

Hierarchical clustering analysis resulted more appropriated to group our data with a two-cluster solution as revealed the internal validation process. Thus, two profiles (clusters) were generated (Figure 2).

It is important to point out that all female applicants (n = 87) were grouped into Cluster 1 along with 178 men while Cluster 2 included male applicants (n = 62) solely. Therefore, sex explained a high percentage of data variation which confirmed our initial hypothesis. This simple matching analysis (natural grouping) of the identified clusters is shown in Table 3 which also shows the characteristics and differences between clusters. Importantly, all measured variables showed significant differences between generated profiles except for age.

## 4. Discussion

The purpose of this cross-sectional study was to profile the body composition of applicants to a PES major at the Universidad Veracruzana, a southeastern Mexican university. We implemented an unsupervised machine learning-based to profile for the first time body composition data in young Mexican freshmen. In this regard, we based our discussion from other similar populations from national or international studies.

Similar to previous research in young adults [29,30,31], males and females that participated in this study showed normal values of BMI. It seems that the nutritional status was appropriate for the specific energy needs of the PES major (which includes heavy physical training). Our sex-dependent analysis showed that females had higher values on whole-body fat (+59.8%) than men as reported previously [32,33,34,35]; nevertheless, women had lower visceral fat content (−34.4%). In the total sample, lower %FM values were found compared to first-year Spanish university students [36]. Similarly, %FM values were lower in both sexes compared to Brazilian PES students [37]. Thus, it should be noted that the characteristics of the participants included in the present study differed from the samples reported in other geographical locations.

The mean %MM of males and females included in our study was lower (8.5 and 18%, respectively) than those reported in Spanish university population [38] although differences in body composition methods should be considered. On the other hand, based on BIA measurements, young female Korean university students have shown lower %MM values than women included in this study [39]. These race and ethnic differences in body composition should be taken into account when proposing intervention activities from university counselors and teachers/lecturers. We also provide percentile tables based on body mass, BMI, and %FM so it can be used to categorize the applicants for PES in similar population.

We were also able to generate two statistically different profiles (Cluster 1 = 265, Cluster 2 = 62) for almost all measured variables (except age). These age-independent clusters represented: (i) applicants of both sexes that were shorter, lighter, with lower adiposity and higher lean mass, and at low risk for metabolic syndrome (lower BMI, MA, and visceral adipose content) (Cluster 1); and, (ii) a group of overweight male applicants with higher whole-body and visceral fat content, taller, with lower %MM and estimated energy expended at rest, and at higher risk for metabolic syndrome (Cluster 2). These results would provide relevant information to universities offering this undergraduate curriculum to identify applicants in which is necessary to increase physical activity levels or make decisions about interventions to avoid potentially risky behaviors while reducing the risk of metabolic syndrome. Pérez-Morales et al. [40] reported that eating disorders are associated with variables such as boredom, sadness and depression besides the accumulation of adipose tissue, increasing overweight and obesity prevalence. In this regard, school managers should consider their monitoring and prevention activities in college students. In fact, García-Alcala et al. [18] reported that the prevalence of obesity tripled while the prevalence of overweight doubled in Mexican adolescents between 1994 and 2008. This is indicative of change towards a sedentary lifestyle and poor nutritional intake. Educational institutions must address this health issue by fostering environments that promote active lifestyles and healthy eating [41].

### Limitations and Future Directions

This study has certain limitations that should be considered when drawing practical inferences and setting future research. First, the analyzes were not controlled for age, race, level of physical activity (certainly many participants were sedentary, others practiced sports). For example, the small proportion of the female population makes it difficult to analyze this subgroup. Second, we did not record physical fitness or academic success variables; thereby, more research is needed to find associations with body composition in this type of student. Third, the estimated values of body composition and BMR should be analyzed carefully given that more accurate results can be obtained with bioelectrical impedance vector analysis (BIVA). Future research might evaluate associations between these morphological features with academic performance, psychometric variables and health-related outcomes throughout the college career.

## 5. Conclusions

Body composition assessment in young applicants to a PES program at southeastern Mexico showed normal adiposity and BMI values. As expected, significant differences on %FM and %MM were found between males and females although the higher visceral fat content in men should be highlighted for future prevention and intervention strategies. For practical purposes, we identified two body composition profiles using an unsupervised machine learning algorithm: (i) Cluster 1 were applicants of both sexes that were shorter, lighter, with lower adiposity and higher lean mass, and at low risk for metabolic syndrome; and, (ii) Cluster 2 was a group of overweight male applicants with higher whole-body and visceral fat content, taller, with lower %MM, and at higher risk for metabolic syndrome. Future research might evaluate associations of applicants’ body composition with academic performance, psychometric variables and health-related outcomes.

## Figures and Tables

**Figure 1 ijerph-19-15685-f001:**
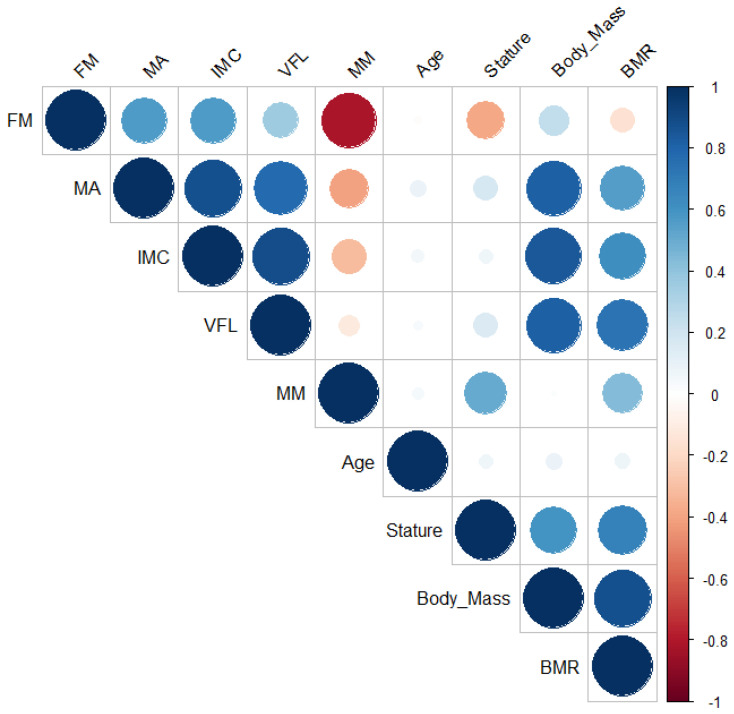
Draftsman correlation plot. Positive correlations are displayed in blue and negative correlations in red color. Color intensity and the size of the circle are proportional to the correlation coefficients.

**Figure 2 ijerph-19-15685-f002:**
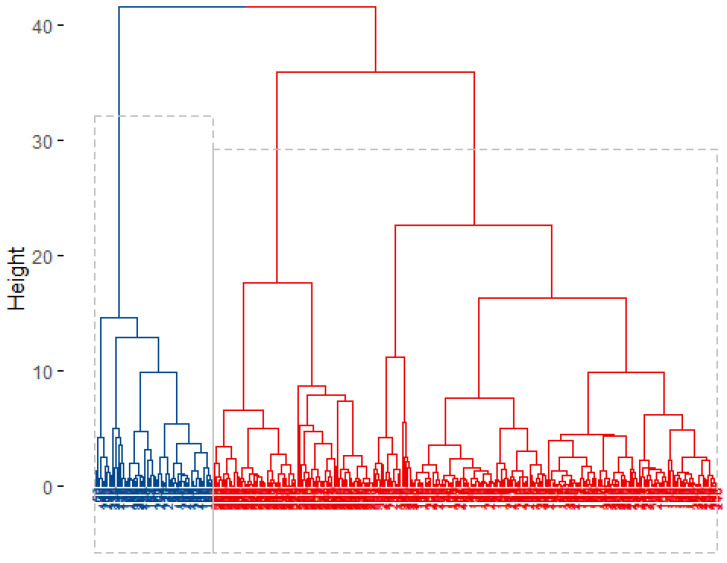
Dendrogram of the bottom-up agglomerative clustering. Each leaf corresponds to one student. Students that are similar to each other are combined into branches, which are themselves fused at a higher height. The height of the fusion, provided on the vertical axis, indicates the (dis)similarity/distance between two profiles/clusters. The hierarchical tree was cut in order to partition the data into clusters (blue for Cluster 1 = 265 and red for Cluster 2 = 62).

**Table 1 ijerph-19-15685-t001:** Descriptive statistics of the undergraduate students by sex.

Variable	Males (n = 240)		Females (n = 87)		*p* Value	η^2^
	Mean (SD)	95% CI	Mean (SD)	95% CI		
Age (years)	18.1 (0.3)	18.0–19.0	18.1 (0.2)	18.0–19.0	0.274	0.003
Stature (cm)	170.7 (7.2)	125.0–193.0	158.9 (5.9)	147.0–179.0	<0.001	0.394
BM (kg)	69.2 (11.9)	67.6–70.6	57.7 (9.3)	55.7–59.7	<0.001	0.185
BMI (kg·m^−2^)	23.7 (3.5)	17.3–37.3	22.9 (3.6)	17.2–33.5	0.054	0.011
FM (%)	20.1 (7.9)	4.8–81.1	33.6 (7.8)	6.8–51.4	<0.001	0.375
MM (%)	40.7 (4.7)	9.6–49.4	27.0 (3.6)	19.7–47.2	<0.001	0.526
MA (years)	28.0 (12.8)	11.6–89.0	28.3 (9.8)	18.0–53.0	0.217	0.004
BMR (kcal)	1654.1 (161.4)	1210.0–2280.0	1263.4 (101.3)	1069.0–1528.0	<0.001	0.549
VFL (cm^3^)	5.8 (3.0)	1.0–16.0	3.8 (1.2)	1.0–7.0	<0.001	0.104

Data are expressed as mean (SD) with the corresponding 95% confidence interval (95% CI). BM = body mass; BMI = body mass index; FM = fat mass; MM = muscle mass; MA = metabolic age; BMR = basal metabolic rate; VFL = visceral fat level. The statistically significant differences at a level of 0.05 for the Mann–Whitney U test are shown. Effect size as eta-squared (η^2^).

**Table 2 ijerph-19-15685-t002:** Percentiles for body mass, BMI and %FM for the participants.

	Percentiles (P)								
	10	20	30	40	50	60	70	80	90
BM (kg)	52.2	56.6	58.8	61.7	64.3	67.8	70.3	75.4	82.7
BMI (kg·m^−2^)	19.4	20.6	21.4	22.3	22.9	23.8	24.6	26.1	28.6
FM (%)	12.4	15.2	17.9	19.7	21.8	24.8	28.3	32.4	37.1

BM = body mass; BMI = body mass index; FM = Fat Mass.

**Table 3 ijerph-19-15685-t003:** Characteristics of the undergraduate students by identified profiles.

Variable	Cluster 1 (n = 265)		Cluster 2 (n = 62)		*p* Value	η^2^
	Mean (SD)	95% CI	Mean (SD)	95% CI		
Sex (n, %)	F (87, 100%)	M (178, 74.1%)	F (0, 0.0%)	M (62, 25.8%)		
Age (years)	18.0 (0.2)	18.0–18.1	18.1 (0.3)	18.0–18.1	0.395	0.002
Stature (cm)	165.8 (8.2)	164.8–166.8	173.2 (7.4)	171.3–175.1	<0.001	0.121
BM (kg)	61.7 (8.2)	60.7–62.7	84.8 (9.4)	82.4–87.2	<0.001	0.425
BMI (kg·m^−2^)	22.3 (2.6)	22.0–22.7	28.1 (3.0)	27.3–28.8	<0.001	0.353
FM (%)	22.6 (10.4)	21.4–23.9	28.0 (5.1)	26.7–29.3	<0.001	0.089
MM (%)	37.2 (8.1)	36.2–38.2	35.8 (3.1)	35.0–36.6	<0.001	0.038
MA (years)	23.8 (7.5)	22.9–24.7	46.1 (10.8)	43.3–48.8	<0.001	0.405
BMR (kcal)	1476.8 (178.9)	1455.1–1498.4	1863.7 (119.8)	1833.3–1894.2	<0.001	0.449
VFL (cm^3^)	4.2 (1.6)	4.0–4.4	9.5 (2.5)	8.9–10.2	<0.001	0.419

Data are expressed as mean (SD) with the corresponding 95% confidence interval (95% CI) unless otherwise is indicated. BM = body mass; BMI = body mass index; FM = Fat Mass; MM = Muscle Mass; MA = Metabolic Age; BMR = Basal Metabolic Rate; VFL = Visceral Fat Level. The statistically significant differences at a level of 0.05 for the Mann–Whitney U test are shown. Effect size as eta-squared (η^2^).

## Data Availability

Any researcher that contacts this project Principal Investigator, L.M.G.-M. (lgomez8@uabc.edu.mx) will have access to the study data, in accordance with the informed consent provided by participants on the use of confidential data.

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
