# Peer review of "Body Composition Profiles of Applicants to a Physical Education and Sports Major in Southeastern Mexico"

_ijerph, 2022, doi:10.3390/ijerph192315685_

Round 1

Reviewer 1 Report

Frankly speaking, even though adopting the convenience sampling and the uneven distribution with gender, as well as a small sample size, this work has also yielded some meaningful and interesting findings by a cross-sectional survey research on body composition profiles of applicants from the major of physical education and sports in southeastern Mexico. Some detailed problems can be revised by authors. For example, at first, the keywords can be changed as follows:body composition profiles; university applicants; physical education and sports major; southeastern Mexico; hierarchical clustering analysis. Secondly, the sign of effect size, namely η2, need be presented with italics. Thirdly, table 1, table 3, and figure 1 has been cross-page. Moreover, some spelling errors are existed in note below table 1 and table 3, namely effect size eta-squared not et-squared. Fourthly, in table 1, table 2, and table 3, the expression of FM and MM are incorrect, they are FM (%) and MM (%), not %FM and %MM. Then, the conclusions should be refined and mainly correspond to the research question, rather than including many detailed content. Lastly, the references should be seriously revised by authors according to corresponding norm of this journal. Thank you!

Author Response

Dear reviewer, 

Thanks for your comments.
Please see our responses attached.

Sincerely, 
The authors

Reviewer 2 Report

Dear authors,

Your submitted paper "Body Composition Profiles of Applicants to a Physical Education and Sports Major in Southeastern Mexico” is beneficial for this Science field and for a view of the body composition of current students in Physical Education and Sports.

However, I recommend making some minor corrections before issuing this paper.

Abstract

Row 40 – key words: Body Composition is stated in the title (It is not necessary to state within the keywords)

Introduction

Row 52-54

The end of the first paragraph does not make sense. You are describing during the paragraph issues that the students face within their studies at the University. However, the last sentence is focused on the necessity to evaluate candidates before they start. Is there a proven connection between the mentioned problems during the study (in this paragraph) and "the demand for energy", as you state in the last sentence? Focus the last sentence on your research in relation to the above variables.

Discussion

Row 216-217

In the part of the discussion, it is better to state the differences in percentages than p-values. Part of the results is intended for this.

Row 216,219,220

You stated a comparison with other studies. Why are these studies not included in part of the introduction? where it is possible to mention and introduce other geographical locations in which a similar approach is carried out.

Conclusions

Row 248-249

It is not clear, how related your results with this sentence. "In general, the nutritional status is appropriate for 248 the specific energy needs of the PES major which includes heavy physical training".

Author Response

(The authors gave the same response as above.)

Reviewer 3 Report

This cross-sectional study aimed to determine the body composition profile of candidates (n = 327) for the Specialization in Physical Education and Sports. For that, a set of variables on body composition were evaluated. Here are my thoughts on the study:

Major

Introduction

1. It is necessary to go deeper into the "justification" for carrying out this study.

a) Presentation of previous studies carried out in Mexico. Certainly, other authors have already evaluated the Nutritional Status of the Mexican population;

b) Therefore, the authors should present the findings of these studies and develop the gaps that exist in the literature;

c) It is also necessary to present the "contribution" of this study to world science. What does this study bring new?

2. At the end of the Introduction, the authors must again present the objectives of the study, and if any, also the hypotheses tested.

Methodology

1. It is necessary to detail how the study was disseminated: posters, radio, etc.

2. There is no detail on the inclusion and exclusion criteria;

3. There are no details on procedures to verify data normality;

Results

1. I consider it a strong point of the study.

Discussion

1. This section could be expanded. There is a wealth of results, however, they have been little explored;

2. Considering that this study has "serious" limitations, the authors should present at least one paragraph to list everything that was not evaluated by this study. For example, the analyzes were not controlled for age, race, level of physical activity (certainly many participants were sedentary, others practiced sports). All these points and many others "still have to be confronted" with the international literature.

Author Response

(The authors gave the same response as above.)

Round 2

Reviewer 3 Report

I congratulate the authors for the corrections and new information. The study was much more complete!